# Rainfall Spatial Estimations: A Review from Spatial Interpolation to Multi-Source Data Merging

**Qingfang Hu [1,\*], Zhe Li [2,3], Leizhi Wang [1], Yong Huang [4], Yintang Wang [1] and Lingjie Li [1]**

[1]   State Key Laboratory of Hydrology, Water Resources and Hydraulic Engineering Science,
     Nanjing Hydraulic Research Institute, Nanjing 210029, China; leizhi668@foxmail.com (L.W.);
     hy121_2000@126.com (Y.H.); ytwang@nhri.cn (Y.W.); ljli@nhri.cn (L.L.)
[2]   Key Laboratory of Terrestrial Water Cycle and Surface Processes, Institute of Geographical Sciences and
     Resources, Chinese Academy of Sciences, Beijing 100101, China; zli875@wisc.edu
[3]   Department of Civil and Environmental Engineering, University of Wisconsin-Madison,
     Madison, WI 53706, USA
[4]   Anhui Provincial Key Laboratory of Atmospheric Science and Satellite Remote Sensing, Anhui Institute of
     Meteorological Sciences, Hefei 230031, China
[\*]   Correspondence: huqf@nhri.com or hqf_work@163.com; Tel.: +86-136-7513-3592

**Abstract:**   Rainfall is one of the most basic meteorological and hydrological elements. Quantitative rainfall estimation has always been a common concern in many fields of research and practice, such as meteorology, hydrology, and environment, as well as being one of the most important research hotspots in various fields nowadays. Due to the development of space observation technology and statistics, progress has been made in rainfall quantitative spatial estimation, which has continuously deepened our understanding of the water cycle across different space-time scales. In light of the information sources used in rainfall spatial estimation, this paper summarized the research progress in traditional spatial interpolation, remote sensing retrieval, atmospheric reanalysis rainfall, and multi-source rainfall merging since 2000. However, because of the extremely complex spatiotemporal variability and physical mechanism of rainfall, it is still quite challenging to obtain rainfall spatial distribution with high quality and resolution. Therefore, we present existing problems that require further exploration, including the improvement of interpolation and merging methods, the comprehensive evaluation of remote sensing, and the reanalysis of rainfall data and in-depth application of non-gauge based rainfall data.

**Keywords:** rainfall; spatial interpolation; radar; satellite; atmospheric reanalysis; rainfall merging

---

## 1. Introduction

Precipitation is one of the most basic meteorological and hydrological elements and has intricate tempo-spatial variability. Accurate information about the precipitation distribution in space is the basis for scientifically understanding global or regional changes in processes involving water and its associated materials and energy, which is of great significance for the promotion of meteorological and hydrological monitoring and forecasting to enhance the capability to cope with natural disasters and optimize water resources management [1–3]. Therefore, spatial estimation of precipitation has been a vital scientific issue of common concern in many fields, such as meteorology, hydrology, ecology, geology, and so on [4–9].

In recent years, numerous efforts on quantitative rainfall spatial estimation have been made. At present, the number of rainfall spatial estimation methods available is relatively high, and new methods are still proposed continuously. These methods are based on special physical and mathematical principles and are adequate for different conditions and tempo-spatial scales. It is

difficult to class the existing methods into different groups. However, from the information source perspective, it appears that the development of rainfall spatial estimation has generally experienced three stages of development. The first stage was spatial interpolation based on the measurements of rain gauges, which was the traditional way to obtain distributed rainfall information. However, the density as well as the spacing required for this method restricts the results obtained by rain gauges. The second stage was remote sensing retrieval and atmospheric model calculation, both of which are areal inherent and can obtain spatial continuous rainfall across a certain region. Since the 1980s, remote sensing retrieval and atmospheric model calculation have gradually become important methods for rainfall spatial estimation. However, due to the influences of remote sensing instruments, atmospheric model capacity, and other factors, their uncertainty is relatively prominent. Since the 1990s, rainfall spatial estimation of precipitation has developed to the third stage, that is, the stage of multi-source data merging. Recently it has become one of the most important hotspots in meteorology and hydrology and can be used to obtain better precipitation information by integrating various kinds of information, such as gauge observations, remote sensing retrieval, and atmospheric reanalysis estimates [10,11].

With the development and interaction of meteorology, hydrology, remote sensing, and spatial statistics, investigation on rainfall spatial estimation has made significant progress in the three stages above and has produced a number of algorithm and rainfall datasets. However, rainfall occurs in a complex process of continuity and intermittence [12,13], with extremely intricate spatial variability. Therefore, accurate acquisition of rainfall spatial distribution still faces a series of challenges. Some authors have reviewed various rainfall spatial estimation methods and the corresponding rainfall datasets [9,14–16]. However, to the best of our knowledge, reviews of the rainfall spatial estimation methodology that completely cover gauge, radar, satellite, atmosphere reanalysis, and multi-source merging are still rare. Hence, in light of the move from spatial interpolation to multi-source merging, this paper aims to summarize the state-of-the-art methods of rainfall spatial estimation and discuss the existing problems. Under this main framework, this paper discusses the recent progress using some of the typical approaches and datasets for each of the three development stages of rainfall estimation. In the following sections, a comprehensive review on rainfall spatial interpolation, remote sensing retrieval, atmosphere reanalysis data, and multi-source data merging is presented. Finally, some issues related to these four aspects are recommended. Considering that spatial variability is virtually the most complex aspect among the meteorological and hydrological variables, this paper aims to provide a valuable reference for improving the quantitative acquisition of rainfall data as well as data on other hydro-meteorological elements, such as air temperature and soil moisture.

## 2. Rainfall Spatial Interpolation

Spatial interpolation is the traditional way to transform point-wise rainfall into areal rainfall. It is a process of tapping and utilizing the spatial autocorrelation of rainfall and spatial intercorrelation between rainfall and related explanatory variables under a specific mathematical framework. Over the years, the evolution of rainfall spatial interpolation has occurred in two directions. One was the introduction of new mathematical statistical theories or methods to improve the utilization of spatial correlation information. Second was the integration of more explanatory variables into the quantitative estimation models to increase the amount of effective information for spatial interpolation. At present, there is no literature summarizing all rainfall spatial interpolation algorithms, but most of the more than 50 interpolation algorithms listed by Li and Andrew [17] have been applied for the spatial estimation of precipitation. For specific rainfall spatial interpolation algorithms, usually, the denser the rain gauge measurements, the higher the estimation accuracy for rainfall and its application, such as hydrological modelling [18–20]. Thus, a minimum density of rain gauge is needed for rainfall spatial estimation. However, when the density of a rain gauge exceeds a certain limit, the estimation accuracy will not change significantly when evaluated by usual statistical indices [21–24]. On the other hand, the effect of introducing auxiliary information on rainfall spatial estimation accuracy depends on



two factors. One is the correlation between auxiliary variables and rainfall—when they are strongly correlated, the introduced auxiliary information will produce a significant effect. Second is the richness of rain gauge measurements. When the density of rain gauge network is low, the marginal effect of introducing auxiliary information is more obvious [25–28]. The accuracy of precipitation spatial interpolation often varies with the study area, time scale, and precipitation type. There is no optimal method for all circumstances, and in some cases, the accuracy of complex methods is not higher than, and may even be lower than, that of simple algorithms [16,21,29–31]. Thus, any rainfall interpolation method has its own merits and demerits. It is highly recommended to select appreciate interpolation methods in terms of the application objective, the geographic and gauge conditions of the study area as well as the temporal and spatial scales.

The spatial interpolation algorithm for precipitation can be classified from different points of view [16,17]. This paper mainly comments on the progress of five methods in rainfall spatial estimation, namely multiple regression, geostatistics, high-accuracy surface modeling, machine learning, and hybrid interpolation. Table 1 lists some corresponding representative literature.

**Table 1.** Summary of representative studies on rainfall spatial interpolation in recent years.

| Sources | Process | Interpolation Methods | Findings |
|---|---|---|---|
| Li (2018) [32] | Compared the rainfall interpolation using GWR and GTWR in the Huaihe River Basin in eastern China. | GWR&GTWR | GTWR can describe the non-stationary spatio-temporal relationship between rainfall and explanatory variables. As the density of rain gauges gradually decreases, the advantage of GTWR over GWR begins to emerge. |
| Hutchinson (1995) [33] | Interpolated annual rainfall for a region of south eastern Australia using Thin plate splines (TPS), a method of GAM | TPS | The main advantage of TPS over competing geostatistical techniques is that splines do not require prior estimation of spatial auto-covariance structure. |
| Adhikary et al. (2017) [34] | In two river basins in Victoria State of Australia, the comparison was made for the performancet of monthly rainfall spatial interpolation using five methods including OCK (Ordinary CK), KED and Radial basis function (RBF). | OK&OCK&KED&IDW&RBF | Among the five methods, OCK with elevation as auxiliary information produced the best estimation accuracy. |
| Ly et al. (2011) [35] | In two mountain basins in Belgium, the influences of different geostatistical methods and theoretical variogram models on daily rainfall spatial estimation accuracy were explored. | OK&UK&OCK&KED&IDW& Thiessen polygons | Estimation accuracy of daily rainfall using UK and OCKwith elevation as auxiliary variables is lower than OK and IDW. |
| Cecinati (2017) [36] | A test on gauge measurement error and its influence on rainfall spatial estimation results through was made in a river basin in southern Netherlands. | OK&OKUD&KED&KEDUD | Considering the error of rain gauge measurements can better describe the uncertainty of rainfall spatial estimation and has some positive effect on improving the prediction results. |
| Chen et al. (2011) [37] | The performance of HASM, IDW, OK and Spline in annual rainfall spatial interpolation in Dongjiang River Basin of South China was compared. | HASM&IDW&OK&Spline | The accuracy of annual rainfall interpolation by HASM is significantly higher than classical methods such as IDW, OK and Spline. |
| Hewitson (2005) [38] | A conditional interpolation method was established to estimate precipitation based on the determination of dry and wet state by self-organizing feature mapping (SOFM), followed by comparison with the classical Cressman interpolation method. | SOFM& Cressman interpolation | The conditional interpolation method based on SOFM can better describe the daily rainfall filed than Cressman method. |
| Guan & Wilson (2005) [39] | With climate and geographical explanatory variables, a hybrid method of Auto-Searched Orographic Atmospheric Effects Detrended Kriging (ASOADeK) was applied to monthly precipitation spatial interpolation in mountainous areas of New Mexico, the United States. | ASOADeK & CK&OK& Parameter Elevation Regression of Independent Slopes Model(PRISM) | ASOADeK more comprehensively reflects the influence of climatic and topographic factors on the spatial variability of precipitation, and the interpolation accuracy of monthly precipitation is higher than OK and CK, and equal to PRISM. |
| Seo et al. (2015) [40] | A hybrid algorithm combining RK, RKNNRK was proposed and compared with other five methods including OK, RK and NNRK. | RKNNRK&RK&NNRK&SK&OK&UK | The accuracy of RKNNRK, RK and NNRK is higher than SK, OK and UK, and RKNNRK ranks the first amongall the six algorithms. |

## 2.1. Multiple Regression

Multivariate regression models can be used to quantitatively estimate the rainfall distribution in space by establishing the linear or nonlinear response relationship between rainfall and explanatory variables. In the early stages, ordinary least squares regression (OLS) was usually used, but the assumption of OLS that precipitation and explanatory variables are correlated in a globally stationary way and that the residuals are subject to independent normal distribution does not comply with the reality [41–43]. As a result, some literature has introduced new regression methods, such as Geographically Weighted Regression (GWR) [44] and the Generalized Additive Model (GAM) [45] into rainfall spatial interpolation. Compared with OLS, these two regression models provide significant improvements in modeling assumption and regression form. They are able to better describe the spatial nonstationary or nonlinear response relationship between rainfall and related influencing variables.

GWR is a variable-parameter spatial regression technique. The key idea of GWR is that the relationship between dependent variables and independent variables is non-stationary in space, and the regression coefficient varies in space. GWR has significantly improved the ability to analyze the variation in spatiotemporal characteristics of precipitation and has attracted wide attention regarding quantitative precipitation estimation as well as other spatial variables [27,41,46,47]. Huang et al. further expanded the spatial variable coefficient regression to provide time-space significance, that is, they established the Geographically and Temporally Weighted Regression (GTWR) which allows adjacent sample locations with positive significance to the regression center in the spatio-temporal neighborhood to be found [48]. Li conducted comparison tests for GWR and GTWR, and demonstrated that with a gradual decrease of the rain gauge network density, improvement due to the introduction of temporal correlation into the precipitation interpolation began to appear [28].

GAM is the semi-parametric extension of the Generalized Linear Model (GLM), and the quantitative relationship between the expectation of the response variable and the smooth function of the prediction variables is established via the link function, so it is feasible to analyze the nonlinear relationship between the response variable and the predictors. The distribution of the response variable in GAM can be any exponential family distribution, and the influences of some independent variables can be described using a nonlinear smooth function, allowing this method to flexibly detect the complex relationships between data. GAM has been successfully applied in rainfall spatial estimation [49–52]. There are several commercial or non-commercial software packages for GAM, probably the most famous of which is ANUSPLINE, which was developed by the Australian National University [33,53].

## 2.2. Geostatistics

Geostatistics is a branch of statistics that explores natural phenomena that are both random and structural. Geostatistics originated from mineral resources evaluation, but it has been widely applied in many fields, such as climate, hydrology, environment, and ecology [54–57]. Geostatistical methods gather many algorithms, including Ordinary Kriging (OK), Universal Kriging (UK), Kriging with External Trend (KED) and Co-Kriging (CK) [58]. Among them, OK only considers the spatial autocorrelations of the predicted variable and assumes that its structural component is locally stationary in space. UK, KED, and CK consider the influence of auxiliary factors on the predicted variable and make use of the spatial autocorrelation of the predicted variable and its cross-correlation information with related auxiliary variables, but they have different modeling methods and calculation processes. UK and KED describe the trend change of the predicted variable in a certain spatial neighborhood by establishing the functional relationship between the predicted variable and explanatory variables. CK describes the spatial correspondence between the predicted variable and explanatory variables through a covariation function. Many studies have compared the rainfall spatial estimation performance of OK, CK, KED, and other methods [25,43,59–61]. Theoretically, due to considering the influence of explanatory variables, CK and KED are better than OK at describing rainfall spatial variability. However, the advantage of these two methods over OK is still subject to a series of factors, such as rainfall type, the correlation strength between rainfall and auxiliary variables, the density of rain gauges, and so on [35].

OK and other geostatistical methods are based on spatial variation functions. Previously, spatial variation functions were generally fitted using parametric methods, such as OLS or weighted least squares (WLS). In the late 1990s, Marcotte [62] and Yao [63] respectively proposed the non-parametric calculation methods for variation functions based on the Fast Fourier Transformation (FFT), and the obtained spatial variation functions were in the form of a two-dimensional matrix. This method has been applied in rainfall spatial estimation [36,64]. Other studies investigated the influence of variation function uncertainty on rainfall spatial interpolation with different approaches. Ly et al. [35] compared the effects of different spatial variation function models on the spatial interpolation results. Plouffe et al. [65] pointed out that the Bayesian Kriging method [66], which takes the uncertainty of spatial variation functions into account, could produce better estimation results on monthly precipitation than OK, spline interpolation, and other methods under certain conditions. Cecinati [36] exploited Kriging for Uncertain Data (KUD) [67] to evaluate the effect of the rain gauge measurement error for different rainfall intensities on spatial estimation and found that OK for Uncertain Data (OKUD) and KED for Uncertain Data (KEDUD) had better performance than conservative OK and KED methods.

### 2.3. High Accuracy Surface Modeling

High Accuracy Surface Modeling (HASM) is a spatial interpolation and prediction method proposed by Yue [68,69] based on the theory of differential geometry. On the basis of the Gauss–Codazii Equation [70], which is satisfied by the first and second fundamental quantities of space surface, HASM transforms the problem of surface simulation into a system of large sparse linear equations with a symmetric positive definite matrix [71]. After several improvements, HASM became increasingly perfect in theory and solved the error problem in classical surface modeling [72]. HASM can also overcome the smoothing effect of spatial interpolation to some extent, and its estimation accuracy is not as sensitive to the distances among sampling points [71]. The computational performance of HASM was also continuously improved. Due to the successive development of the adaptive algorithm [73], the pretreatment conjugate gradient algorithm [74], and the multi-grid algorithm [69], the iterative solution efficiency for large linear equations is raised for HASM.

As a relatively new spatial statistical method, the application of HASM is not as extensive as multiple regression and geostatistics, but there have been many successful examples of the spatial interpolation of elements such as topography, soil, temperature, and precipitation [72,75–77]. Numerical experiments and examples have shown that the estimation accuracy of HASM is better than that of OK and other classical methods. For example, Chen et al. [37] pointed out that the estimation effect of HASM on annual precipitation in the Dongjiang River Basin of China was significantly better than the three classical algorithms of Inverse Distance Weighting (IDW), OK, and Spline. Zhao and Yue [78] studied the spatial interpolation of perennial average precipitation in mainland China and came to a similar conclusion. In addition, HASM has been applied in rainfall spatial downscaling [77,79]. The main disadvantage of HASM is that it cannot directly process spatial element interpolation with obvious trend components.

### 2.4. Machine Learning

Multiple regression, geostatistics, and other methods need to make some assumptions about the variability of spatial variables and their relationships with the associated explanatory variables, but some assumptions are difficult to satisfy. The machine learning algorithm, on the other hand, is based on the idea of using data-driven analysis to explore the relationship between relevant variables. Its estimation results only depend on the grey box or black box relationship established by sample training between the input and output. It does not need to use definite mathematical formulas and has a strong ability to handle nonlinear relations. Hence, the machine learning algorithm is effective for rainfall spatial estimation with its complex influencing factors and vague physical mechanism. At present, machine learning algorithms such as the artificial neural network (ANN) [34,38,80,81],

association rule mining [82], fuzzy inference [83], and random forest [84] have been used in rainfall spatial interpolation and have had some success.

*2.5. Hybrid Interpolation*

Hybrid interpolation algorithms improve the precision of rainfall spatial estimation by integrating different kinds of algorithm. Generally, hybrid interpolation first uses one method to preliminarily estimate the predicted variables and then uses another interpolation algorithm to calculate the residuals of the former algorithm. Finally, the estimation results of the two algorithms are synthesized. The precision of a hybrid algorithm is generally higher than that of a single constituent algorithm. A common type of hybrid algorithm is the coupling of regression and geostatistics. First proposed by Bénichou and Le Breton in 1987, AURELHY (Analysis Using the Relief for Hydrometeorology) is a hybrid interpolation method that combines multiple regression and Kriging methods. Guan [39] established a residual Kriging method based on optimal window regression and applied it to spatial variability analysis and quantitative estimation of precipitation in mountainous areas. The Regression-Kriging (RK), proposed by Hengl [85], combines generalized least squares regression and OK, and its modeling is more flexible than KED and CK. RK has numerous applications in precipitation spatial interpolation [30,61,86]. Sun et al. [87] proposed a hybrid interpolation algorithm that combines GWR and OK. In addition, some scholars built hybrid interpolation algorithms by integrating the machine learning algorithm and geostatistics. For example, Seo et al. [40] established a hybrid algorithm combining regression Kriging and neural network residual Kriging (RKNNRK). It can be seen from examples that the accuracy of RKNNRK is not only higher than RK and UK, but it is also higher than that of the common neural network and residual Kriging coupled algorithm (NNRK). Zhang et al. [88] used the support vector machine (SVM) to explore the nonlinear relationship between precipitation and terrain factors and further used OK or IDW to estimate the residuals of SVM.

## 3. Remote Sensing Rainfall Retrieval

*3.1. Rainfall Estimation with Radar*

Weather radar is a ground-based active microwave remote sensing technology, which uses the backscattering characteristics and echo intensity of cloud and rain particles from electromagnetic waves to monitor the instantaneous rainfall intensity within the scanning range, and it can dynamically track the rainfall process in three dimensions. As an important method of quantitative precipitation estimation that is inherently areal, the development of radar rainfall estimation technology mainly depends on two aspects: one is the continuous application of high-performance radar instruments; the other is the continuous improvement of radar rainfall retrieval algorithms based on a single radar and radar network [89,90].

Regarding the radar instrument, great effort has been made to develop the dual polarization radar (DPR) and phased array radar (PAR) [91–93]. DPR is used in practical applications at present. It can emit horizontal and vertical linear polarization signals simultaneously. Compared with the single polarization radar (SPR), DPR can obtain more parameters of cloud and rain particles, thus improving the performance of echo recognition, precipitation type diagnosis, cloud and rain microphysical process detection, and other aspects, which has important significance for improving the detection effect of precipitation [90,94]. PAR can quickly and accurately convert the detection beam to complete full spatial scanning in one minute [95]. PAR scanning is characterized by higher spatial and temporal resolution and less blind areas than SPR, which strengthens its ability to detect and track the rapid evolution of the meso-micro scale weather system [96,97]. However, PAR is still in the stage of scientific experimentation [98]. The development of radar equipment is also reflected in the selection of wavebands. As the long-band is less affected by rainfall attenuation and backward scattering phase shift in the detection of heavy rainfall, S-band and C-band radars are usually used in operational observation [99,100], while the X-band radar, which is heavily affected by rainfall attenuation, was less

used in the past. However, the X-band radar features a low transmission power and small antenna diameter and can sensitively detect weak meteorological targets, and the rain attenuation problem has been continuously alleviated. Therefore, recently, the X-band radar has gradually become an important tool to strengthen the local rainfall detection ability, in some sense making up for the insufficient coverage of the S band and C band radars [101].

Radar precipitation retrieval involves two processes: radar observation data quality control and radar echo–rainfall conversion. In the phase of quality control, it is necessary to first calibrate the radar instrument to correct the working parameters as much as possible. At the same time, it is necessary to eliminate or reduce the negative impact on the quality of radar base data caused by a series of factors from the observation environment and observation object variation. For observation environment variation, these factors include the abnormal propagation and blockage of the radar beam, the non-precipitation echo, and so on. For observation object variation, these factors include missing rain clusters, non-uniform beam filling, electromagnetic wave signal attenuation, and the bright band of the melting layer [102]. The influence of radar beam blockage could be solved by means of hybrid scanning with different elevation angles [103] and dual-polarization monitoring [104]. The problem of abnormal high reflectivity of the radar caused by the bright band of the melting layer is usually treated with vertical reflectivity profile correction [105].

For radar reflectivity and rainfall intensity conversion, the usual practice is to establish an empirical Z-R regression equation based on the radar meteorological equation and raindrop size distribution with the reflectivity-rainfall data from the same time period and location. The Z-R regression equation is usually in the form of a power exponential, but, in fact, it is difficult to describe the extremely complex radar reflectivity-rainfall intensity correspondence with power exponential equations. Therefore, a probability fitting technique (PFT) based on the relationship with the measured rainfall-reflectivity frequency distribution was developed later, which does not rely on samples from the same location and time period to establish a regression relationship. PFT includes a number of forms such as ordinary PFT [106], window PFT [107], and window correlation PFT [108]. In addition, Hasan et al. [109] proposed a method to establish the Z-R relationship based on conditional probability estimation of the kernel density. In general, there has been in-depth, international research on the conversion relationship between radar reflectivity and rainfall, but the Z-R relationship is often significantly different depending on the location, season, and precipitation type, so the resulting errors in radar precipitation retrieval are still prominent [103].

*3.2. Satellite Rainfall Retrieval*

Satellite-borne sensors can detect rainfall information in a larger coverage area than ground-based radars and have the advantage of applicability to special situations, such as oceans, large lakes, high mountains, and deserts. Therefore, satellite rainfall retrieval is of great significance in large-scale climatic and hydrological research. Depending on differences in source data, rainfall retrieval algorithms can generally be classified into four types: visible light and infrared (VIS/IR), passive microwave (PMW), active microwave (AMW), and multi-sensor precipitation estimation (MPE).

The VIS/IR retrieval algorithm estimates the surface rainfall by establishing statistical relationships between rainfall intensity and cloud field parameters such as cloud type, cloud area, cloud top bright temperature detected by optical sensors aboard geostationary satellites [110,111]. The VIS/IR algorithm can obtain continuous rainfall intensity information, but as characteristic cloud field information, such as the cloud top light temperature, is not directly related to rainfall, the accuracy of the VIS/IR algorithm is rather low. PMW algorithms are based on detecting information from a microwave radiometer carried by a polar-orbiting satellite. Because microwaves can probe rainfall information inside the cloud, PMW algorithms are more direct and effective than VIS/IR algorithm. PMW retrieval algorithms can be roughly divided into empirical methods [112], semi-empirical methods [113], physical model methods [114], and physical profile methods [115]. Furthermore, the satellite-borne precipitation radar, an active satellite microwave sensor, has overcome not only the

defect that the optical sensor cannot penetrate atmospheric cloud and rain, but also the demerit of passive microwave sensor that it cannot provide vertical structure information about precipitation. TRMM PR, the world's first satellite-borne precipitation radar, greatly advanced the development of the AMW precipitation retrieval algorithm. The standard PR algorithm estimates the real radar reflectivity through the vertical profile of satellite-borne radar reflectivity and then calculates the precipitation rate [116]. The core satellite of the Global Precipitation Mission (GPM) is equipped with a dual-frequency precipitation radar (DPR), which can estimate rainfall more accurately than TRMM PR, a single frequency radar. In particular, DPR can raise the identification ability of micro precipitation and solid precipitation in the cold season [117,118]. However, due to the Earth System Model's method of observing polar-orbiting satellites, neither passive nor active microwave retrieval can obtain continuous rainfall intensity information.

VIS/IR or MW information based rainfall retrieval methods have their respective advantages and disadvantages, so the MPE algorithms, which integrate both of them, have become the main method of satellite rainfall retrieval. The MPE algorithms are classified into calibration methods and cloud trail methods [119], and most MPE algorithms applied nowadays belong to the former. The basic idea of calibration methods is to establish the empirical relationship between GEO-IR and MW and then estimate rainfall rate using the corrected IR. The algorithms of the Global Precipitation Climatology Project (GPCP) [120] and TRMM Multi-satellite Precipitation Analysis (TMPA) [121] both belong to calibration method. Cloud trail methods are based on the cloud motion vector interpolation PMW information obtained by IR and obtain the precipitation rate over a large range of space, and its representative algorithms are the Climate Prediction Center morphing technique (CMORPH) [122] and Global Satellite Mapping of Precipitation (GSMaP) [123]. The development of MPE algorithms and their rainfall datasets is closely linked with satellite and satellite-borne sensors. Before 1997, the MPE data sources were mainly GEO-IR and SSM/I data provided by satellites of the Defense Meteorological Satellite Program (DMSP). During this period, the Adjusted GOES Precipitation Index (AGPI) and other algorithms were developed and rainfall datasets with coarse spatial resolution (2.5° × 2.5°), such as GPCP, were established. After 1997, TMI, PR, AMSR-E, and AMSU-B sensors carried by TRMM, NOAA, and EOS satellites provided much richer microwave information and produced rainfall datasets such as TMMM with high spatial resolution (2.5° × 2.5°). Since 2014, with the in-orbit operation of GPM satellites and DPR, Integrated Multi-satellite Retrievals for GPM (IMERG) was proposed, which can calibrate and integrate all of the microwave and infrared information from GPM satellite groups and other satellites, and theoretically, has a higher precision for instantaneous precipitation estimation. Table 2 lists the basic information of representative global or quasi-global satellite precipitation datasets since the launch of TRMM. Datasets in this table are pure satellite remote sensing retrieval data, without calibration by surface gauge data.

**Table 2.** Basic information of representative global or quasi-global satellite rainfall datasets.

| Short Name | Full Name | Data Sources | Resolution and Frequency | Spatial Coverage | Period | Latency | Reference |
|---|---|---|---|---|---|---|---|
| TMPA 3B42-RT | TRMM Multi-satellite Precipitation Analysis (TMPA) 3B42 Real Time | TMI, TCI, SSM/I, SSMIS, AMSR-E, AMSU-B, MHS, GEO IR | 0.25°/3 h | 50° S–50° N | 1998–2015 | 9 h | Huffman et al. (2007) [121] |
| CMORPH | CPC MORPHing technique | TMI, AMSR-E, AMSR-2, SSM/I, SSMIS, AMSU-B, MHS | 0.25°/3 h, 8 km/30 min | 60° S–60° N | 1998–present | 18 h | Joyce et al. (2004) [122] |
| GSMaP-MVK | Global Satellite Mapping of Precipitation Moving Vector with Kalman | GMI, TMI, AMSR-E, AMSR2, SSM/I, SSMIS, and MHS/AMSU-A | 0.10°/1 h | 60° S–60° N | 2000–present | 2–3 days | Ushio et al. (2009) [123] |
| GSMaP-NRT | Global Satellite Mapping of Precipitation Near Real Time | GMI, TMI, AMSRE, AMSR-2, SSM/I, SSMIS, and MHS/AMSU-A | 0.01°/1 h | 60° S–60° N | 2007–present | 4 h | Kubota et al. (2007) [123] |
| PERSIANN | Precipitation Estimation from Remotely Sensed Information using Artificial Neural Networks | Meteosat, GOES, GMS, SSM/I, polar/near polar precipitation radar, TMI, AMSR | 0.25°/6 h | 60° S–60° N | 2000–present | 2 days | Sorooshian et al. (2000) [124] |
| PERSIANN-CCS | Precipitation Estimation from Remotely Sensed Information using Artificial Neural Networks (PERSIANN) Cloud Classification System | Meteosat, GOES, GMS, SSM/I, polar/near polar precipitation radar, TMI, AMSR | 0.04°/30 min | 60° S–60° N | 2003–present | 1 h | Hong et al. (2004) [6] |
| IMERGE | Integrated Multi-satellite Retrievals for GPM (IMERG) | GMI, AMSR-2, SSMIS, Madaras, MHS, ATMS | 0.10°/30 min, 3 h, 1 d | 60° S–60° N | 2014–present | 4 h | Hou et al. (2014) [117] |

The performance of satellite precipitation datasets has been widely evaluated on different spatial and temporal scales. The World Meteorological Organization (WMO) implemented the Program to Evaluate High Resolution Precipitation Products (PEHRPP), to carry out a dynamic evaluation and comparison of precipitation datasets such as TRMM and CMORPH at global and continental scales [119,125]. Maggioni et al. [126] summarized the accuracy evaluation and verification results of high-resolution satellite precipitation data from various continents and oceans in the TRMM era. Beck et al. [11] assessed the performance of 23 rainfall datasets at a global scale using surface rainfall records from 76,086 gauges. Sun et al. [9] compared the differences of 30 global rainfall datasets, including 12 kinds of satellite precipitation data across various space-time scales, and explored the opportunities and challenges for the future development of global precipitation data. These studies show that satellite rainfall data obtained by different retrieval algorithms are often significantly different, but there is no single type of data that ubiquitously has the best performance. IR/VIS retrieval-based datasets often miss or underestimate light rainfall and topographic rainfall, while the MW retrieval based datasets, although generally better than IR/VIS based datasets, have obvious deficiencies in the estimation of topographic rainfall, especially rainfall in the cold season [127]. Error characteristics of certain satellite rainfall datasets are also obviously different across different climatic and geographical backgrounds and change across different space-time scales. Compared with non-humid areas, satellite rainfall is generally more reliable in humid areas. Additionally, satellite performance in mountainous areas with high complex terrain is worse than in flat and open areas. The quality of satellite precipitation data is also affected by the surface. In large inland waters, the quantity and magnitude of precipitation events is usually overestimated by satellite datasets. The accuracy of satellite rainfall data has obvious seasonal differences—it is relatively high in the rainy season and relatively low in the dry season. It should be noted that the error of the satellite rainfall dataset is also related to the type and magnitude of precipitation. For various satellite rainfall datasets, there is a relatively large deviation in the identification of snow and rain-snow mixed precipitation, and the accuracy generally decreases with the increase of precipitation intensity. Turk et al. [119,125] pointed out that the error in MPE data stems from two aspects: one is the estimation error of microwave data on instantaneous precipitation; the other is the cumulative error of retrieval algorithms on the evolution process of precipitation. With the advent of the GPM era, the evaluation of IMEGE data has become a hot topic. IMERG data are considered to be the best satellite rainfall data available at present, and their ability to detect extreme heavy precipitation and solid precipitation has been somewhat improved as compared with the TMPA dataset [126,128,129]. However, there are also studies indicating that the improvement of IMERG is not as obvious as that of TMPA and that it is even slightly lower than TMPA 3B42RTV7 [130,131]. In general, the error characteristics and influencing factors of pure satellite retrieval rainfall data are extremely complex, and it is still difficult to directly apply them to meteorological and hydrological practices.

## 4. Atmosphere Reanalysis Rainfall Data

### 4.1. Main Reanalysis Datasets

Atmosphere reanalysis has been used internationally to restore the long-term historical climate records since the late 1980s. Using data assimilation technology, this methodology combines numerical weather forecast results with measurements from the surface gauge network, radar, and satellites. Thus, atmosphere reanalysis is the integration of irregular observation data and regular output of atmospheric numerical model under a unified physical dynamic framework. Atmosphere reanalysis can provide information on climatic elements with space coherency and time continuity including long series of precipitation data, which is extremely important for meteorological and hydrological investigation across different time and space scales.

Over the last 40 years, the development of reanalysis datasets has experienced four generations, each with improved quality and diversity. In this process, several global atmosphere reanalysis

datasets with significant international influence have been released by the United States, the European Union, and Japan. Table 3 lists the basic information of representative atmosphere reanalysis precipitation datasets. The National Center for Environment Predication (NCEP) and National Center for Atmospheric Research (NCAR) of the United States developed NCEP/NCAR Reanalysis 1 based on 3D variational assimilation technology using data series since 1948 in the highest time resolution of 6 h and a space resolution of $2.5° \times 2.5°$ [132]. The NCEP and the Department of Energy (DOE) further proposed the improved version of NCEP/NCAR Reanalysis 1—NCEP/DOE Reanalysis 2 [133], using data series from 1979 up to now, and the space resolution increased to $0.5° \times 0.5°$. NCEP also developed NCEP-CFSR data with a maximum time resolution of up to 1 h and a space resolution of $0.3125° \times 0.3125°$ for a data series from 1979 up to now. Currently, CFSR has been updated to version 2— CFS V2 [134] with a space resolution of $0.20° \times 0.20°$, but it has not been computed back to 2011 and earlier. NASA has also developed the Modern-Era Retrospective Analysis for Research and Application (MERRA) dataset [135], a set of global reanalysis data, but its time and space resolution is lower than that of NCEP-CFSR. The JRA-55 dataset [136], developed by the Japan Meteorological Agency (JMA), has a maximum time resolution of 3 h and a space resolution of $0.5625° \times 0.5625°$, with the data series tracing back to 1958. The European Centre for Medium-Range Weather Forecasts (ECMWF) developed ERA-Interim data using the ECMWF comprehensive forecast system and 4D variation assimilation technology, covering a period from 1979 to the present. ERA-Interim is the early data from ERA-70, which will be released in the future. In July 2017, ERA ReAnalysis 5(ERA-5), the latest version of ERA reanalysis data, was released, and it is expected to replace ERA-Interim [137]. The first batch of ERA5 is just available for the period of 2010–2016, but by early 2019, it will be extended to the period from 1950 to the present. Compared to ERA-Interim, ERA-5 uses one of the most recent versions of the Earth System Model's data assimilation methods applied at ECMWF, which means it refines the parametrization of the Earth's processes. The temporal and spatial resolutions of ERA-5 are both improved compared with those of ERA-Interim: from 6-hourly in ERA-Interim to hourly in ERA-5 and from $0.75° \times 0.75°$ in the horizontal dimension and 60 levels in the vertical direction in ERA-Interim to $0.25° \times 0.25°$ and 137 levels in ERA-5.

**Table 3.** Basic information of representative atmosphere reanalysis rainfall datasets.

| Short Name | Full Name | Assimilation Schemes | Resolution and Frequency | Spatial Coverage | Period | Reference |
|---|---|---|---|---|---|---|
| NCEP/NCAR Reanalysis 1 | The National Center for Environment Predication (NCEP) and National Center for Atmospheric Research (NCAR) Reanalysis 1 | 3D-Var (Spectral statistical interpolation) | $2.5° \times 2.5°$/6 h | Global | 1948–present | Kalnay et al. (1996) [132] |
| NCEP/DOE Reanalysis 2 | The NCEP and the Department of Energy (DOE) Reanalysis 2 | 3D-Var | $0.5° \times 0.5°$/6 h | Global | 1979–present | Kanamitsu et al. (2002) [133] |
| NCEP-CFSR | National Centers for Environmental Prediction(NCEP) Climate Forecast System Reanalysis | 3D-Var | $0.2° \times 0.2°$/1 h | Global | 2012–present | Saha et al. (2014) [134] |
| MERRA | Modern-Era Retrospective Analysis for Research and Application system | 3D-Var | $0.5° \times 0.67°$/1 d | Global | 1979–present | Rienecker et al. (2011) [135] |
| JRA-55 | Japanese 55 year ReAnalysis | 4D-Var | $0.5625° \times 0.5625°$/3 h | Global | 1958–present | Ebita et al. (2011) [136] |
| ERA-Interim | European Centre for Medium-range WeatherForecasts ReAnalysis Interim | 4D-Var | $0.75° \times 0.75°$/6 h | Global | 1979–present | Dee et al. (2011) [137] |
| ERA5 | European Centre for Medium-range WeatherForecasts ReAnalysis 5 | 4D-Var | $0.25° \times 0.25°$/1 h | Global | 2000–present | Hersbach & Dee (2016) [137] |

*4.2. Assessment and Comparison*

With the improvement of tempo-spatial resolution, more and more attention has been paid to the value of precipitation reanalysis data in climate and hydrology analysis. The ability of atmosphere reanalysis precipitation data to represent the decadal variability in precipitation on a global and large regional scale has been confirmed. Prakash et al. [138] found that the characteristics of global and regional inter-annual and inter-decadal water changes reflected in the three types of reanalysis precipitation data, including ERA-Interim, were similar to the two types of satellite precipitation data, including CPCP, after calibration with surface rain gauge data. Lin et al. [139] pointed out that five types of reanalysis precipitation data, including NCEP-CFSR and ERA-Interim, were all able to represent the actual global monsoon precipitation wave from 1979 to 2011 well, in which ERA-Interim performed the best comparatively. Chen et al. [140], after comparing satellite and surface precipitation observation data, pointed out that, on the whole, JRA-55, NCEP-CFSR, ERA-Interim and MERRA data have the ability to represent diurnal changes in warm season precipitation in East Asia, and Huang [141] presented a similar view. Tesfaye et al. [142], based on ground gauge measurements over 33 years, verified that the four types of reanalysis data, including ERA-Interim, can reflect the overall diurnal precipitation changes in Ethiopia. Sun et al. [9] pointed out that the reanalysis precipitation data is less accurate than the satellite precipitation data, but Worqlul et al. [143] found that the performance of CFSR data in the upper reaches of the Nile River was better than that of TRMM 3B42 data and could better simulate the runoff process. Although ERA5 precipitation was only released very recently, its improvement over ERA-Interim and other global precipitation datasets has attracted significant attention. Hénin et al. [144] assessed the ability of ERA5, ERA-Interim, and TRMM 3B42RT and TRMM 3B42 to reflect daily extreme precipitation events over the Iberian Peninsula over the period 2000–2008. It was found that ERA5 reanalysis gave large improvements over ERA-Interim, and it also outperformed the two satellite-based datasets. For the conterminous US during 2008–2017, the daily performance of ERA5 ranked first among the 15 satellite or reanalysis precipitation datasets without gauge adjustment [145]. In addition, there are studies appraising the effect of driving land surface process simulation using ERA5 with respect to ERA-Interim. Albergel et al. [146] found that when the Soil, Biosphere, and Atmosphere model (ISBA) was forced by ERA-5 data, it obtained a consistent improvement in surface field simulation over ERA-Interim. This was particularly evident for the land surface variables linked to the terrestrial hydrological cycle. Wang et al. [147] concluded that differences in the precipitation fields of ERA5 with respect to ERA-Interim have a larger influence on the sea ice evolution than the 2 m air temperature over the arctic sea ice.

The above results highlight the importance of rainfall information provided by atmosphere reanalysis. By using models to relate and combine information from diverse observations, reanalysis arguably offers the potential to obtain rainfall fields with long series. However, uncertainty is still evident, and the spatial resolution is coarse even for ERA5. Quantifying uncertainty in reanalysis datasets remains an important challenge for increasing their utility. Greater evaluation is the key to increasing confidence. At the same time, it should be stressed that the differences in numerical forecast models, meteorological observations, and assimilation systems result in performance diversity among various types of reanalysis precipitation data, and the same data may have a remarkable simulation ability at the decadal, interannual, and annual scales. Since most satellite remote sensing data were produced after 1979, the quality of reanalysis data before the 1970s was poor, so special caution should be taken in the application of reanalysis data of this period.

## 5. Multi-Source Rainfall Merging

Different methods of precipitation observation or estimation have different abilities to reflect the rainfall state in space, with varied reliability and error characteristics. Theoretically, a rain gauge network can only obtain pointwise rainfall information, and its ability is limited by the density and spacing of the gauges, while remote sensing and reanalysis precipitation feature strong spatial continuity and wide coverage but with prominent local errors, making it difficult to directly apply them

in practice currently. Therefore, to estimate the real state of rainfall in a better way, trials can be made under a certain optimization criterion to integrate information from different sources with different levels of spatial and temporal resolution and precision, and this is the basic idea of multi-source rainfall merging. The input information of rainfall merging can include ground surface observations, radar and satellite retrieval, and atmosphere reanalysis rainfall as well as other relevant auxiliary variables. Rainfall merging began in the 1970s with the associated precipitation estimation combining radar and rain gauge measurements [148]. With the emergence of global satellite and atmosphere reanalysis precipitation datasets since the late 1990s, the research on gauge—satellite or gauge—reanalysis rainfall merging has become increasingly active [149–154].

*5.1. Rainfall Merging Algorithms*

At present, numerous rainfall merging algorithms have been proposed. In this paper, these algorithms are roughly classified into three categories, which include the initial field correction mode, the interpolation mode with auxiliary information, and the optimal matching mode. Table 4 shows the 11 representative algorithms of the three modes, their basic principles, technical characteristics, and references. More than one method listed in Table 4 is based on the spatial interpolation algorithm, reflecting the close link between spatial interpolation and rainfall merging in regard to mathematical meaning. From the perspective of precipitation information, Table 4 not only involves the merging of gauge and radar, satellite, or reanalysis precipitation but also the merging of multiple satellite or reanalysis precipitation data. As different types of rainfall data often vary in spatial and temporal resolution, rainfall merging involves not only the matching of rainfall amounts, but also matching on the time-space scales. However, most current merging algorithms focus on the former. Multi-source rainfall merging is a process of integrating different types of rainfall information and a process of blending and matching different errors. A given rainfall merging method, while providing precipitation analysis results, usually produces indices of uncertainty in the estimated results.

We used Category I to denote the initial field correction mode. For this type of algorithm, the first step is to use one or several types of rainfall data to construct a rough initial field similar to the prior information. Subsequently, other kinds of rainfall and auxiliary variables are employed to correct the initial field under a certain optimization criterion, such as the minimum estimated error variance. Finally, the rainfall analysis field is obtained, which can be viewed as a posteriori information and represents the true state of the rainfall field. Representative algorithms of Category I include objective analysis (OA) [155], optimal interpolation (OI) [78], Bayesian filtering [156,157], and scale recursive estimation [158]. Among them, OA is a classical and empirical algorithm. OI generally uses remote sensing or reanalysis data to construct the initial rainfall field. The analyzed rainfall value at a certain location is equal to the sum of the initial value and the deviation value, while the deviation value is calculated by the optimal weighted average of the deviation of several adjacent locations with observations. Of course, for Category I merging methods, the initial rainfall field can also be constructed with the gauge measurements and corrected using radar or satellite rainfall information, such as with Bayesian filtering. Scale recursive estimation (SRE) is a multi-scale rainfall merging algorithm that combines precipitation information and spatial scale conversion. SRE integrates both Kalman filtering and the random cascade model and includes two processes: upward filtering and downward smoothing.

**Table 4.** Basic principles and features of main rainfall merging algorithms.

| Category | Method | Basic Principle | Technical Feature | Reference |
|---|---|---|---|---|
| Category I | OA | Generally, the initial rainfall field is generated with remote sensing or reanalysis rainfall data, and then gradually corrected with the weighted average of the difference between the surface observation value and the initial value in a certain spatial neighborhood. | OA is an empirical local correction method, not taking into account the measurement error of surface observation. The correction weights are decided subjectively, and the analyzed rainfall field obtained is not the optimal estimation result. | Boushaki et al. (2009) [155] |
| | OI | The initial rainfall field is corrected with the weighted average of the difference between the surface observation and the initial value in a certain spatial neighborhood.The optimal correction weights are obtained based on the criterion of minimizing error variance. | OI is also a local optimal estimation method, avoiding the subjectivity of weight selection. Error of the observation field and background field and their spatial correlation need to be inferred in advance. The error variance estimation of the analyzed field can be given. | Shen et al. (2014) [78] |
| | BF | Some kinds of rainfall were used to derive the prior distribution, and others were used to derive the likelihood function. The prior distribution was updated by the Bayesian formula to obtain the posterior probability density distribution of rainfall. The expected value of the posterior probability was taken as the analyzed result. | BF provides a probabilistic analysis framework for multi-source rainfall merging. Analytical solutions of the posterior probability density distribution can be given just for normal distribution. BF can provide the indices fore describing uncertainty of the analyzed results. | Verdin (2015) [157] |
| | SRE | Under the framework of random cascade model, the conversion of precipitation across different scales is realized by the two processes of upward filtering and downward smoothing. | Combining spatial scale conversion and rainfall amount matching, rainfall fields across different scales can be obtained. SRE can merge two or more rainfall data and provide the error measurement of estimation uncertainty. | Gorenburg et al. (2001) [158] |
| Category II | CK | Usually, ground measured rainfall is regarded as the main variable, and remote sensing or reanalysis rainfall as the auxiliary variables. After obtaining the co-variation function between the main and the auxiliary variables, CK equations are used for estimation. | When there are many auxiliary variables, the computation of covariance function is intensive. With strong correlation between the main and auxiliary variables, good estimation results can be obtained. It provides the measure of estimation uncertainty by CK variance. | Velasco-Forero et al. (2009) [64] |
| | KED | Remote sensing, reanalysis rainfall or other auxiliary variables are used to describe the local variation trend of rainfall. The effect of space trend components on the estimated values is reflected by the constraint conditions of the KED equations. | It is necessary to determine the spatial variation function of the residual components, but the residual variation function is coupled with the estimation of local trend components, so an iterative method or other special treatment is needed to solve the KED equations. | Cecinati (2017) [36] |
| | GAM | Rainfall estimation results are the sum of smooth spline function and trend components. The trend are usually described by linear regression of remote sensing, reanalysis rainfall or other auxiliary information. Predicted values are obtained by minimizing the objective function including error square and spline function roughness. | GAM assumes that the error mean is zero and the error variance is stationary in space. The trend components are normally expressed as the global linear regression of covariates. It can provide the uncertainty measure index of the spatial estimation results of precipitation. | Huang et al. (2016) [141] |
| | GTWR | It is a local variable coefficient regression model and extends the spatial estimation of rainfall to 3 D space-time domain. The spatial-temporal correlation between precipitation and related influences is viewed as nonstationary through the spatial variable coefficients. | GTWR can flexibly describe the non-stationary spatial-temporal relationship between the true rainfall and its explanatory variables such as remote sensing precipitation, reanalysis precipitation, and geographical and topographic factors. Besides rainfall amount, error variance of estimation results is also provided. | Li (2018) [28] |
| Category III | BMA | BMA takes any precipitation data as a possible member of the real precipitation state ensemble and measure the importance of each member by the posterior probability density. The weighted average of all the member is finally calculated as the analysis result. | BMA could blend precipitation data more than two kinds and provide the measurement for estimation uncertainty by the posterior variance. The key of this method is the iterative solution of weight coefficient, and usually the solution method of expectation maximization is adopted. | Ma et al. (2018) [32] |
| | PDM | The analyzed rainfall are expressed as the weighted average of different original rainfall data, so that its probability density distribution has the maximum overlap with the original data. | Strict theoretical assumption is lacking for PDM. Iterative process is required to search the weighted for various original data. PDM does not directly provide uncertainty measurement indexes for spatial estimation. | Hasan et al. (2016) [109] |
| | VA | By minimizing the cost function between the analyzed and initial or observed rainfall fields, the optimal estimation results in functional sense is sought. The cost function is usually the weighted average sum of the distance between the analyzed field and the initial field or observation field. | VA is a global optimization method, and can not only the minimize the distance between the analyzed and initial or observation rainfall field, but also include other specific objectives into the cost function. It is necessary to estimate the covariance function of the observed field error and the initial field error in advance and solve it with numerical method. VA does not provide directly the uncertainty measurement indexes for the estimation results. | Li et al. (2015) [156] |

Category II is an interpolation mode that includes auxiliary information. This category takes gauge measurements as the main variables while remote sensing and reanalysis rainfall information are viewed as auxiliary variables. Then, the true rainfall is estimated within the framework of a certain spatial interpolation method. CK, KED, GAM, and GWR methods all belong to this category. CK needs to establish a co-variation function for remote sensing or reanalysis rainfall with gauge measurements. The influences of different types of precipitation information on estimation results are reflected by the CK weight. KED and RK methods take auxiliary precipitation information as independent covariables to deduce the local or global trend components of precipitation in space, while the residual components are estimated using the Kriging equation. In the GAM model, remote sensing or reanalysis of precipitation also exists in the form of independent covariates, but the specific estimation principle is different from that of KED or RK. GWR uses spatial variable coefficients to integrate various precipitation information and quantitatively describe the nonstationary spatial relationships between surface rainfall and auxiliary variables.

Category III is the optimal matching mode, which combines or matches different types of precipitation information with certain optimization criteria and objectives. Typical algorithms of Category III include BMA [32,159], probability density matching [109], the variation method [160,161], and so on. Among them, BMA regards any kind of precipitation as a possible estimation of the real state under the view of ensemble estimation and uses the posterior probability to measure the importance of different estimation results, and finally, takes the weighted average of different estimation results as the analyzed result. Probability density matching seeks a set of optimal weights so that the probability density distribution corresponding to the analyzed results and source rainfall data has maximum overlap. The variation method is a global optimal merging method. By establishing the cost function between the analyzed field and the initial field or observed field, the optimal estimation result in functional analysis sense is directly sought. The variation method generally takes the weighted distance between the analyzed field and the initial field or observed field as the cost function and can also include other specific constraint conditions or objectives into the cost function.

In addition to the methods shown in Table 3, there are some other rainfall merging algorithms. For example, Ehret [162] proposed a conditional merging method. For this method, the OK interpolation results of ground measurements and radar rainfall are respectively used as the precipitation trend component and error component, and the superposition of the two is used as the analysis result. Based on the idea of image processing, Kalinga [163] used wavelet decomposition and reconstruction to realize the merging of radar and gauge rainfall.

## 5.2. Evaluation of the Merging Effect

In order to illustrate the gain of rainfall merging, it is necessary to identify whether the merged rainfall data are of higher accuracy than any original data. A large number of studies have confirmed that the accuracy of remote sensing and reanalysis rainfall is significantly improved after their combination with surface gauge rainfall data. For example, Tian et al. [164] pointed out that even if TRMM 3B42RT, or CMORPH is merged with relatively sparse rain gauge data, the error could be reduced by 47% to 63%. Other studies have pointed out that the merging of ground rainfall with remote sensing or reanalysis rainfall can actually obtain better estimation results than any of the original data. For example, Pan et al. [165] conducted a Bayesian merging experiment using a gauge, radar, and satellite rainfall and stated that the accuracy of three types of source-merged data was better than any single source data. Nie et al. [166] used independent data to verify the OI merging effect of gauge–satellite–reanalysis daily precipitation data on a global scale. This study showed that, relative to the three data sources, the merged precipitation data showed improvement in the spatial pattern, time variation characteristics, and the quantitative and categorical description ability of rainfall events. However, other studies pointed out that, for gauge-based rainfall, the gain achieved from its merging with remote sensing or reanalysis rainfall was significant only when the gauge density was relatively low. Rozante [153] and Woldemeskel et al. [167] respectively pointed out that the net gain of combining

TRMM rainfall with gauge rainfall in South America and Australia was discerned mainly in areas with sparse rain gauges. Li [28] carried out experiments to merge satellite and reanalysis rainfall at monthly and daily time scales with the ground rainfall of different gauge densities and found that the gain of precipitation merging relative to the spatial interpolation of gauge measurements was gradually shown only when the density of rainfall gauge was below a certain threshold. Lu [168] indicated that when the density of the rain gauge was high, the results of radar-gauge merging were similar to those of gauge-based interpolation. These investigations indicated that the noise of pure remote sensing and reanalysis rainfall is rather prominent, and the effective precipitation information they can provide is still relatively limited in essence, so it is more suitable for them to supplement the ground observations under the condition of insufficient rain gauges. The quality and spatial resolution of remote sensing or reanalysis rainfall also affect the merged results. Pan et al. [165] found that the marginal effect of merging gauge rainfall with radar rainfall is more significant than that of CMORPH, and the latter only led to improvements in sparse gauge areas not covered by radar. Chen et al. [169] stated that for the merging of gauge rainfall with TRMM 3B43V7, remarkable improvement occurred when TRMM 3B43V7 data downscaled from $0.25° \times 0.25°$ to 1 km $\times$ 1 km. Of course, the rainfall merging effect is also affected by algorithms. Nerini et al. [170] confirmed the difference in estimation accuracy among various merging algorithms and pointed out that although Bayesian merging and other complex algorithms are relatively perfect in theory, their actual effect may not be better than simple ones, because it is difficult to meet the required assumptions. Nanding et al. [171] compared radar–gauge rainfall merging methods, such as KED and KRE, and found that the KED produced a better effect and was less affected by the density and configuration of rain gauges. McKee et al. [172], Rodriguez et al. [173], and Fadhel et al. [174] all stated that radar–gauge rainfall merging results were simultaneously decided by multiple factors such as the quality of radar rainfall, the distribution of rain gauges, the merging algorithm, and the precipitation type.

Research to explore the hydrological gain of rainfall merging has also become increasingly popular. Both Looper et al. [175] and McKee et al. [176] reported that gauge–radar rainfall merging could raise the precision of runoff forecast in various aspects such as the efficiency coefficient, flood peak flow, and peak occurrence time, but the specific effect changed with the rain gauge density, precipitation intensity, and type. Wang et al. [77] proposed a Bayesian merging method taking into account the singularity of the rainfall field. Using this method to merge gauge and radar rainfall increases the likelihood of catching local heavy rainfall, thus improving the ability to simulate flood peaks in the urban runoff process. Considerable literature on the hydrological effects of gauge–satellite or reanalysis rainfall merging also exists. With relatively sparse rain gauges, the accuracy of runoff simulation is significantly improved after the surface rainfall is merged with satellite or reanalysis data [11,98,154,170]. However, when the density of rain gauges exceeds a threshold value, the marginal effect of gauge and remote sensing rainfall merging is not quite significant [124,177]. Due to the complementarity errors of different remote sensing and reanalysis rainfall techniques, runoff simulation can be improved through their merging. Gebregiorgis et al. [178] proposed the merging method of TMPA 3B42RT, CMORPH, and PERSIANN based on prior information from runoff and soil moisture content simulations, which improved rainfall and runoff estimation. Jiang et al. [159] used the BMA method to integrate multiple satellite precipitation data and found that the runoff simulation effect was better than the use of any single type of precipitation data. The Multi-Source Weighted-Ensemble Precipitation (MSWEP) method [11], which merges gauge observations and several kinds of satellite and reanalysis rainfall data, outperforms the gauge-corrected TRMM dataset TMPA 3B42 in 1058 sparsely gauged catchments globally.

## 6. Conclusions and Future Remarks

A comprehensive review of related literature showed that, thanks to the development of space observation technology and mathematical statistics, quantitative rainfall spatial estimation has greatly improved, which has deepened our understanding of water cycle laws at different space-time scales.

In summary, the method of rainfall spatial estimation has transformed from traditional interpolation to multi-source merging. Rainfall estimation approaches and rainfall datasets have both become increasingly diverse. However, due to the extremely complex spatiotemporal variability and physical mechanism of rainfall, it is still quite challenging to obtain accurate high-resolution rainfall information in space. With the increasing demands of scientific research and practice for high quality rainfall data, there is still a lot of room for theoretical and methodological exploration. Here, the following four issues are especially recommended:

1.  The development of rainfall spatial algorithms: This is required to improve the quantitative description of rainfall spatial variability from new perspectives. In the past, the semi-variance function was used to describe the structure of spatial correlations of rainfall. Recently, some scholars described the spatial correlations of rainfall from the perspective of multiple joint probability distribution and established a quantitative estimation model based on the Copula function. [179]. Second, the spatial interpolation method for short-time scale precipitation should be improved by taking into account the probability of precipitation events. At daily and sub-daily time scales, rainfall has obvious spatial discontinuity, so the distribution of rainy and non-rainy areas needs to be reasonably delineated in the rainfall interpolation process. Thornton [180] and Hewitson [38] explored this issue and proposed a two-stage or conditional estimation method to estimate the precipitation based on the precipitation occurrence probability. The third aspect is to improve the way auxiliary information with uncertainty is used and to develop soft spatial interpolation methods. A lot of auxiliary precipitation information appears in the form of soft data. The scientific use of spatial soft data to improve the estimation effect of precipitation will be an important direction in the future. In addition, another important point is the transition from purely spatial estimation to tempo-spatial interpolation. Kyriakidis and Journel [181] annotated space–time models under a geostatistical framework. Due to the time-space anisotropy of rainfall, it is difficult to directly construct space-time coupled models. However, studies have investigated approaches including additional time information in rainfall spatial estimation. For example, the Meteorological Interpolation Based on Surface Homogenized Data Basis [182], which was developed at the Hungarian Meteorological Service, is a time-space interpolation method that uses climatological information from long time-series to optimize the statistical parameters in geostatistical models.

2.  Extensive evaluation of remote sensing and reanalysis rainfall data: Remote sensing and atmosphere reanalysis rainfall are inherently areal with remarkable uncertainty. Numerous evaluations of previous remote sensing and reanalysis rainfall data have provided an important basis for the correct use of them and improvement of the retrieval algorithm. With the continuous emergence of new global rainfall data, it is necessary to improve the evaluation method and deepen the understanding of various data error characteristics and influencing factors under different climatic and geographical backgrounds and space-time scales. Moreover, most of the existing assessments were highly dependent on using the surface rainfall data as a benchmark, resulting in difficulty directly recognizing the error features of remote sensing and reanalysis precipitation data in areas where ground measurements are insufficient or lacking. Thus, it is necessary to develop new methods to assess remote sensing or reanalysis precipitation data in the absence of sufficient surface observation data. Some studies have made useful attempts in this regard. For example, the triple collocation method (TC method) [183] can get rid of the dependence on surface rainfall observation data. Based on the error relationships among three independent types of remote sensing or reanalysis precipitation data, TC could calculate the correlation coefficient between them and the true rainfall.

3.  Improvement of rainfall merging algorithms: Multi-source rainfall merging is a process of integrating different types of precipitation information as well as balancing and matching the errors. Thus, the understanding of all types of precipitation data error, including the surface observation data, should be deepened. Then, we could refine the methods and models that can

simultaneously merge multiple types of precipitation data and effectively integrate auxiliary information influencing the precipitation distribution. In addition, the spatial and temporal resolution of different types of rainfall data is often significantly different, so multi-source rainfall merging actually includes spatial and temporal scale matching and numerical blending of different types of rainfall data. Previous investigations have shown that the accuracy of satellite precipitation data can be improved by merging with ground rainfall after spatial downscaling [169]. However, most of the current precipitation merging algorithms focus on the blending of precipitation with significant simplification in the matching of space-time scales, and this may affect the actual merging effect. In the future, it is necessary to strengthen the spatio-temporal scale conversion method for precipitation data and better combine scale matching and numerical blending in the merging model. Similar to rainfall spatial interpolation, the number of rainfall merging methods is also high and is increasing. To understand the advantages and disadvantages of various methods, more evaluations and intercomparisons are required. At the same time, we strongly recommend the selection and use of rainfall merging methods according to the conditions of the study area, data sources, and application objectives.

4. In-depth application of non-gauge based rainfall data: At present, non-gauge-based rainfall data, such as remote sensing, reanalysis estimates, and multi-source merging data, have found some experimental applications [6,172,184]. With continuous improvements in accuracy, spatio-temporal resolution, and extension of data length, the potential value of these rainfall data should be further explored. In particular, intensification of their application in hydrology forecasting, water resources management, and drought disaster pre-warning under the condition of insufficient ground observation data is required.

Finally, we emphasize that all of the rainfall spatial estimation methods mentioned above are upward or top-down methods. This derives rainfall data through inverting the atmospheric signals that are reflected or radiated by atmospheric hydrometeors, and it is the most common method of obtaining rainfall data. Recently, some authors have investigated the derivation of rainfall using bottom-up methods by performing hydrology methods backwards. In general, downward methods are based on the water balance equation and derive rainfall using hydrological variable measurements of soil water storage, evapotranspiration, discharge, and glaciers [185–189]. Although downward methods make simplifications to a relatively large degree and some of them cannot measure the distribution of rainfall, they actually provide new ways of breaking down rainfall estimation and are thus worthy of deeper exploration in the future.

**Author Contributions:** investigation, Q.H.; resources, Z.L.; writing—original draft preparation, Q.H.; Z.L.; writing—review and editing, Q.H., L.W.; supervision, Y.H.; project administration, Y.W.; funding acquisition, Y.W.; L.L.

**Funding:** This research was funded by The National Key Research and Development Program of China (grant number: 2016YFC0400902; 2016YFC0400910); National Natural Science Foundation of China (grant number: 51479118); Consulting and Research Program of the Chinese Academy of Engineering (grant number: 2015-ZD-07-02) and Public Welfare Industry Scientific Research Special Fund of the Ministry of Water Resources (grant number: 201501014). The authors also appreciated the constructive comments of the three anonymous reviewers for improving the quality of this paper.

**Conflicts of Interest:** The authors declare no conflict of interest.

**Abbreviations**

The following abbreviations are used in this manuscript:

| | |
|---|---|
| AGPI | Adjusted GOES Precipitation Index |
| AMW | Active microwave |
| AMSR-E | Advanced Microwave Scanning Radiometer for the Earth Observing System |
| AMSU-B | The Advanced Microwave Sounding Unit |
| ATMS | Advanced Technology Microwave Sounder |
| BF | Bayesian filter |
| BMA | Bayesian model averaging |
| BME | Bayesian Maximum Entropy |
| CFSR | Climate Forest System Reanalysis system |
| CK | Co-Kriging |
| CMORPH | Climate Prediction Center morphing technique |
| DMSP | Defense Meteorological Satellite Program |
| DOE | Department of Energy |
| DPR | Dual-frequency Precipitation Radar |
| ECMWF | European Centre for Medium-Range Weather Forecasts |
| EOS | Earth Observing System |
| ERA | European Centre for Medium-Range Weather Forecasts reanalysis systems |
| FFT | Fast Fourier Transformationt |
| GAM | Generalized Additive Model |
| GEO-IR | Geostationary Infrared |
| GLM | Generalized Linear Model |
| GMI | GPM Microwave Imager |
| GMS | GEO Meteorological Satellite |
| GOES | Geostationary Operational Environmental Satellites |
| GPCP | Global Precipitation Climatology Project |
| GPM | Global Precipitation Measurement |
| GSMaP | Global Satellite Mapping of Precipitation |
| GTWR | Geographically and Temporally Weighted Regression |
| GWR | Geographically Weighted Regression |
| HASM | High Accuracy Surface Modeling |
| IDW | Inverse Distance Weighting |
| IMERG | Integrated multi-satellite retrievals for GPM |
| JRA-55 | Japanese 55-year Reanalysis |
| KED | Kriging with External Trend |
| KEDUD | Kriging with External Trend for Uncertain Data |
| KUD | Kriging for Uncertain Data |
| MERRA | Modern-Era Retrospective Analysis for Research and Application system |
| MHS | The Microwave Humidity Sounders |
| MPE | Multi-sensor Precipitation Estimation |
| MSWEP | The Multi-SourceWeighted-Ensemble Precipitation |
| MW | Microwave |
| NASA | National Aeronautics and Space Administration |
| NCAR | National Center for Atmospheric Research |
| NCEP | National Center for Environment Predication |
| NOAA | National Oceanic and Atmospheric Administration |
| OA | Objective analysis |
| OI | Optimum Interpolation |
| OK | Ordinary Kriging |
| OLS | Ordinary Least Square |
| PAR | Phased Array Radar |
| PDM | Probability density matching |

| PEHRPP | Program to Evaluate High Resolution Precipitation Products |
| PERSIANN | Precipitation Estimation from Remotely Sensed Information using Artificial Neural Networks |
| PERSIANN-CCS | PERSIANN Cloud Classification System |
| PMW | Passive microwave |
| RK | Regression-Kriging |
| SRE | Scale recursive estimation |
| SSM/I | Special Sensor Microwave/Imager |
| SSMIS | Special Sensor Microwave Imager Sounder |
| SVM | Support Vector Machine |
| TC | Triple collocation |
| TCI | TRMM Combined Instrument |
| TMI | TRMM Microwave Imager |
| TMPA | TRMM Multi-Satellite Precipitation Analysis |
| TMPA 3B42-RT | TRMM Multi-satellite Precipitation Analysis (TMPA) 3B42 Real Time |
| TRMM | Tropical Rainfall Measuring Mission |
| UK | Universal Kriging |
| VA | Variation Anlysis |
| VIS/IR | Visible/infrared |
| WLS | Weighted Least Square |
| WMO | World Meteorological Organization |

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
