# Peer review of "Rainfall Spatial Estimations: A Review from Spatial Interpolation to Multi-Source Data Merging"

_water, doi:10.3390/w11030579_

Round 1
Reviewer 1 Report
In Introduction part: line 25-29. There are only two stages. The third one is not mentioned. This part needs more elaborations.
Introduction part should rewrite to consider the relevance of different rainfall data source as well as their different applications.
I think the structure of the manuscript should be improved. Any interpolation method has its own capability and it is highly depended to the purpose of the study. For example in flood management studies and water resources studies we have different objectives in terms of rainfall temporal and spatial scales.
Maybe you need to look at different applications of rainfall interpolation such as rainfall runoff modelling, water balance assessment or even raingauge network augmentations and etc...
Furthermore, in field of reanalysis rainfall data, there are lots of peer reviewed papers. It not as simple as you review and comparing them with other source of rainfall data.
Traditional Interpolation methods transform point rainfall into areal rainfall. On the other hands, the RS based rainfall are inherently areal.
Some of the conclusion remarks are pretty obvious.
This work sound a precious one but still needs to be improved.
I just suggest to take a look at the following papers as well:
Assessing the impacts of raingauge density on the simulation results of a hydrological model.
Assessment of rain-gauge networks using a probabilistic GIS based approach.
Effect of the accuracy of spatial rainfall information on the modeling of water,sediment, and NO3-N loads at the watershed level.
Evaluation of reanalysis rainfall estimates over Ethiopia
Author Response
We appreciated the constructive comments from you and have addressed each of your concerns.

Reviewer 2 Report
Comments and Suggestions for Authors
The paper deals with the review of approaches and methods employed in estimating spatially distributed precipitation, based on in-situ observations, remotely sensed / satellite based or modelled precipitation amounts, through utilising interpolation merging or extraction techniques.
Although I do like this kind of review studies, I have several doubts that should be clarified by the Authors before considering the paper for publication. Below are my comments:
General comments:
1. The English quality of the manuscript should be revised and improved. Although most of the text is reasonably well written and well structured, there are some particular (and important) grammatical errors and typos as well as ambiguity in some sentences which may need reconsideration of wording or structure. (some of these points and sentences are indicated in the next section)
2. The Introduction section does not provide sufficient account of previous studies on similar topic. Similarly, the authors also need to establish the novelty of this study in comparison to other such studies.
3. The manuscript may need addition of some further contents. Such as “advantages and disadvantages of the different methods under different conditions i.e. different input data, regions, specific terrains, time scale, spatial scale and so on” and possibly “a brief list of global / regional precipitation data products available based on the methods mentioned in this study.
4. The list of methods does not include some specially developed methods for meteorological interpolation such as MISH (ref.-5), AURELHY (ref.-6) or methods relying on Kirchner’s Methodology / reverse hydrology (references 1-4).
5. Check for consistency of terms such as rainfall /precipitation,
6. There should be a thorough revision and editing of the text-English to make it easier to understand by the reader, especially in the “Abstract”, “Introduction” and “Conclusion” parts. Some of the sentences that need correction or reconsidering are as follows:
· 17,
· 30-31,
· 39-41,
· 43(“In” should start with a small letter; what is meant by “the information kinds”),
· 45,
· 502,
· 523-526,
· 576
References
1. Kirchner, J.W. Catchments as simple dynamical systems: Catchment characterization, rainfall-runoff modeling, and doing hydrology backward. Water Resour. Res. 2009, 45, 2135.
2. Khan, A.J.; Koch, M. Correction and Informed Regionalization of Precipitation Data in a High Mountainous Region (Upper Indus Basin) and Its Effect on SWAT-Modelled Discharge. Water 2018, 10, 1557.
3. Krier, R.; Matgen, P.; Goergen, K.; Pfister, L.; Hoffmann, L.; Kirchner, J.W.; Uhlenbrook, S.; Savenije, H.H.G. Inferring catchment precipitation by doing hydrology backward: A test in 24 small and mesoscale catchments in Luxembourg. Water Resour. Res. 2012, 48, 225.
4. Immerzeel, W.W.; Wanders, N.; Lutz, A.F.; Shea, J.M.; Bierkens, M.F.P. Reconciling high-altitude precipitation in the upper Indus basin with glacier mass balances and runoff. Hydrol. Earth Syst. Sci. 2015, 19, 4673–4687.
5. Szentimrey, T.; Bihari, Z.; Szalai, S. Meteorological Interpolation Based on Surface Homogenized Data Basis (MISH); European Geosciences Union, General Assembly: Vienna, Austria, 2005; Available online: https://www.snap.uaf.edu/attachments/Interpolation_methods_for_climate_data.pdf
http://r-forge.r-project.org/projects/aurelhy/
Author Response
We appreciated your constructive comments and have addressed your concerns, thank you!

Reviewer 3 Report
see attached file.

Author Response
We appreciated the constructive comments and have addressed your concerns, thank you!

Round 2
Reviewer 1 Report
I just think the title can be improved. For example, my suggestion would be:
Rainfall Spatial Estimations: A Review from Spatial Interpolation to Multi-source Data Merging
Author Response
Thank you so much for your advices in improving our manuscript.
The new title seems reasonable, we would love to adopt the new title.
Thank you again!
Best wishes.
Reviewer 2 Report
The manuscript have undergone enough improvements with all the observations and suggestions addressed.
Author Response
We appreciate all the advices from you to improve the quality of our manuscript.
Thank you again and best wishes!
This manuscript is a resubmission of an earlier submission. The following is a list of the peer review reports and author responses from that submission.